# Bmp4 Synexpression Gene, *Sizzled,* Transcription Is Collectively Modulated by Smad1 and Ventx1.1/Ventx2.1 in Early *Xenopus* Embryos

**DOI:** 10.3390/ijms232113335

**Published:** 2022-11-01

**Authors:** Zia Ur Rehman, Faryal Tayyaba, Unjoo Lee, Jaebong Kim

**Affiliations:** 1Department of Biochemistry, Institute of Cell Differentiation and Aging, College of Medicine, Hallym University, Chuncheon 24252, Gangwon-Do, Korea; 2Department of Electrical Engineering, Hallym University, Chuncheon 24252, Gangwon-Do, Korea

**Keywords:** Szl, Bmp4 synexpression, Smad1, BRE, Ventx1.1, Ventx2.1, VRE, transcriptional regulation, *Xenopus laevis*

## Abstract

Sizzled (Szl) is a secreted frizzled protein, having a sequence homology with the extracellular cysteine-rich domain (CRD) of the Wnt receptor, ‘Frizzled’. Contrary to the other secreted frizzled like proteins (Sfrps), *szl* belongs to the bone morphogenetic protein 4 (Bmp4) synexpression group and is tightly coexpressed with Bmp4. What is not known is how the *szl* transcription achieves its Bmp4 synexpression pattern. To address the molecular details of *szl* transcription control, we cloned a promoter of size 1566 base pairs for *szl* (bps) from the *Xenopus laevis* genomic DNA. Luciferase and eGFP reporter gene results of this *szl* promoter (−1566 bp) in its activation and repression patterns by Bmp4/Smad1 and a dominant negative Bmp4 receptor (DNBR) were similar to those of the endogenous *szl* expression. Reporter gene assays and site-directed mutagenesis of the *szl* promoter mapped an active Bmp4/Smad1 response element (BRE) and a cis-acting element, which competitively share a direct binding site for Ventx1.1 and Ventx2.1 (a Ventx response element, VRE). *Smad1* and *ventx2.1* alone increased *szl* promoter activity; in addition, the binding of each protein component was enhanced with their coexpression. Interestingly, Ventx1.1 repressed this reporter gene activity; however, Ventx1.1 and Ventx2.1 together positively regulated the *szl* promoter activity. From our analysis, Ventx2.1 binding was enhanced by Ventx1.1, but Ventx1.1 inhibitory binding was inhibited by co-injection of Ventx2.1 for the VRE site. The inhibitory Ventx1.1 co-injection decreased Smad1 binding on the *szl* promoter. In a triple combination of overexpressed Smad1/Ventx1.1/Ventx2.1, the reduced binding of Smad1 from Ventx1.1 was recovered to that of the Smad1/Ventx2 combination. Collectively, this study provides evidence of Bmp4/Smad1 signaling for a primary immediate early response and its two oppositely behaving target transcription factors, Ventx1.1 and Ventx2.1, for a secondary response, as they together upregulate the *szl* promoter’s activity to achieve *szl* expression in a Bmp4 synexpression manner.

## 1. Introduction

Vertebrate embryogenesis has evolved from a common ancient prototype, emerging around 600 million years ago [1], and all vertebrates share common genes and pathways for early embryogenesis as it is believed to be a recapitulation of a previous common evolutionary track [2]. One of the selected genes in early embryogenesis of both vertebrates and invertebrates is Bmp4 (Dpp) [3]. Bmp4 has an evolutionally conserved signaling that uses the common strategy of fine-tuning its signal intensity via extracellular and intracellular signal effectors, affecting its target transcriptome [4]. In vertebrate embryogenesis, Bmp4 is an early morphogen, which patterns the dorso-ventral embryonic axis and is specifically required for ectoderm and ventral cell fate specification [5,6,7], and loss of Bmp4 signaling leads to expansion of neural tissues [8,9]. Ventral regions of both ectoderm and mesoderm are the major sites of Bmp4 function, in which Bmp4 binds to its receptor to phosphorylate and to activate the receptor-related Smads (Smad1/5/8) [10]. Upon activation by Bmp4, receptor-associated Smads make a heterotetrameric complex with Smad4 and translocate to the nucleus, targeting various sets of genes by binding directly to Bmp4 response elements found in promoters of Bmp4 target genes; these are known as Bmp4 synexpression genes [11,12]. These set of genes include various subgroups from the *bambi*, *szl*, *ventx, gata*, *id* and *msx* families [13]. For the Bmp4 synexpression group, however, a mechanistic study is required on the transcriptional regulation for synexpression, especially with respect to the promoter of a given target gene.

Sizzled (Szl) belongs to the family of secreted soluble signaling antagonistic proteins (Sfrps) that are structurally related to the membrane bound Wnt receptor, Frizzled [14]. Different from other Sfrps, *szl* is a Bmp4 synexpression gene and its levels are dependent on the spatial distribution of the Bmp4 gradient, defined as a high concentration in the ventral region, transitioning to a lower concentration in the dorsal region [15]. The function of Szl has been reported as being a negative feedback regulator of Bmp4 signaling via indirect stabilization of the Chordin protein [16]. For the *Xenopus* embryo, research has demonstrated that Szl is initially abundant in the animal cap and the ventral marginal zone at the mid-blastula transition stage and during/after gastrulation, with its functional target, *chordin,* being almost completely repressed [17]. There have been previous reports of Bmp4/Smad1 signaling directly controlling the *szl* expression via a probable Smad1 binding element [18,19,20]. However, the presence of Smad1 binding and its response elements within the *szl* promoter have not been reported, although transcriptional regulatory sites for other Bmp4 target genes, including the *ventx* family members, have been described [21,22]. In addition, for Bmp4 synexpression of *szl* transcription, involvement of a secondary response for other Bmp4 target synexpression genes, including *ventx* family genes, has not been addressed. Here, we performed a detailed *szl* promoter analysis for presence of a Smad1 binding cis-acing element (BRE) as well as a Ventx1.1/2.1 response element (VRE).

In the present study, we isolated the *szl* promoter containing Bmp4-signaling regulatory cis-acting elements. The isolated promoter of 1566 bps in size had a transcription pattern that was similar to that of the endogenous *szl* in gastrula of *Xenopus* embryos. We identified for the first time that the isolated promoter contained a Smad response cis-acting element (BRE) and a Ventx response element (VRE), modulating *szl* transcription. Additionally, we found that Ventx1.1 alone suppressed *szl* expression and its promoter activity; however, Ventx2.1 alone had the opposite effect of increasing the relevant expression and promoter activity. Each Ventx1.1 and Ventx2.1 alone recognized the same Vent response element (VRE) located at the proximal region of *szl* promoter. Interestingly, the combination of both *ventx1.1* and *ventx2.1* overexpression enhanced *szl* promoter activity more than that of *ventx2.1* overexpression alone. Inhibitory Ventx1.1 coexpression enhanced activatory Ventx2.1 binding on the VRE site. Ventx2.1 also reduced inhibitory Ventx1.1 binding on the same VRE, leading to overall stimulation of *szl* promoter activity. In addition, inhibitory Ventx1.1 reduced Smad1 binding on the BRE site. The triple combination of Smad1/Ventx1.1/Ventx2.1 recovered the Ventx1.1 mediated reduced binding of Smad1 to the level of Smad1/Ventx2.1 combination. Collectively, these results demonstrate that *szl* promoter contains direct binding cis-acting elements of Smad1, Ventx2.1 and Ventx1.1 for Bmp4 synexpression, which together positively modulate *szl* transcription in gastrula *Xenopus* embryos.

## 2. Results

### 2.1. Isolated Szl(-1566) Promoter Contains the Bmp4 Regulatory Elements Required for Bmp4 Synexpression

Bmp4 is the dominant signaling of the ventral side of embryo initiated from early onset of mid-blastula transition stage [23], and overexpression of Bmp4 profoundly influences the ventral marker genes and downregulates the dorsal genes. Loss of Bmp4 signaling results in upregulated expression of neural genes [24]. The Bmp4 gradient defines the ectoderm and neuroectoderm of the embryo [25]. Reported data suggest that *szl* is a Bmp4 synexpression gene and its transcription is dependent on the Bmp4 gradient, indicating that *szl* transcripts accumulate in ectoderm and in the ventral lateral marginal zone [13,26,27]. We analyzed the spatio-temporal expression of Bmp4 synexpression genes including *szl*, *ventx1.1* and *ventx2.1* through RT-PCR analysis. We found that *szl* transcription start from stage 8 at the ventral side of the embryo where Bmp4 signaling is abundant (Appendix A Appendix A). To examine the effects of Bmp4 signaling on *szl* endogenous expression, *bmp4* (1 ng/embryo), *dnbr* (1 ng/embryo, *dominant negative bmp4 type Ia receptor*) or *smad1* (1 ng/embryo) mRNAs were individually injected to the animal pole region of one-cell stage embryos. The qRT-PCR analysis of Bmp4 target genes including *szl*, *ventx1.1*, *ventx2.1*, *gata2* (a Bmp regulated gene, which is expressed in ventral mesoderm and required for normal hematopoiesis [28]), and *xbra* was performed with the isolated RNA from the embryos or animal cap explants (AC) at stage 11. The isolated RNA samples were balanced with ornithine decarboxylase (*odc*), a house-keeping gene which is used for equal loading control in RT-PCR experiments. As expected, Bmp4 and its intracellular signal mediator Smad1 overexpression increased *szl* transcription of the resultant AC (Figure 1a, graph 1). Conversely, the Bmp signaling inhibition reduced the *szl* mRNA level by treating embryos with *dnbr* (a dominant negative Bmp-type 1a receptor) mRNA. The dominant negative Bmp-type 1a receptor, which lacks the intracellular kinase domain [29,30], inhibits the Bmpr1a-mediated Bmp signaling pathways in vivo and promotes complete neuralization [15,31,32,33]. The other ventral marker genes, such as *ventx1.1*, *ventx2.1* and *gata2* behaved similarly to the *szl* expression pattern. Bmp/Smad1 overexpression also enhanced *xbra*, a pan-mesodermal marker [34,35] which was turned out as a secondary Bmp4 response gene in the present study (via CHX experiment; Appendix A). Xbra also plays a cooperative role in the transcription of Bmp target gene *ventx1.1* via a physical interaction with Smad1 [5,36].

To examine the transcriptional regulation of *szl*, a 1566 bp 5′-flanking region of *szl.L* from translational start site (TLS) was cloned into *pGL3-luc+* (Figure 1b) and *pGL3-eGFP+* reporter gene vectors (Figure 1d). The cloned *szl(-1566)* promoter construct was injected (40 pg/embryo) to the one-cell stage embryos, and the reporter assays at the designated stages were performed as shown in Figure 1c,e. The relative luciferase activities indicated that the cloned *szl(-1566)-luc+* promoter activity was enhanced by Bmp4 (Figure 1c) and the pattern of eGFP expression mimicked the pattern of endogenous *szl* expression (Figure 1e, upper panels). The eGFP signal was dramatically reduced by *dnbr* co-expression with Bmp4 signaling inhibition (Figure 1e, bottom panels). The relative luciferase (Luc) activities maximally responded to Bmp4 treatment at stage 12 (Figure 1c). The reporter gene (eGFP) expression was localized to the ventral half (Figure 1e, bottom side of embryo at stage 10.5) of the embryo and was at its maximum in stage 10.5 and stage 12. Bmp4 response and the eGFP expression levels then gradually were reduced by post stage 12 (Figure 1c,e). Reported research has indicated that *bmp7.1* and *bambi* are also Bmp4 synexpression genes [13]. We then cloned and tested the reporter gene activities of *pGL3.luc+(bmp7.1:-3143)* (Appendix A, Appendix A), *pGL3.luc+(bambi:-3073)* (Appendix A, Appendix A) and *pGL3.eGFP+(bambi:-3073)* (Appendix A, Appendix A) promoters. Both of these promoter luciferase activities and their eGFP fluorescence intensities were reduced in the presence of *dnbr* mRNA similarly to those of the *szl* promoter. We next examined the luciferase activities and eGFP signal intensities at different developmental stages to determine whether the *szl* promoter activity with Bmp4 treatment is replicating its endogenous transcriptional expression patterns. *Szl* was expressed around the ventral marginal zone (the lower part of the stage 10 embryo). Data from RNA expression patterns (Appendix A), eGFP imaging (Figure 1e) and the reporter gene activities of *szl* in various developmental stages (Figure 1c) indeed indicated a similar pattern of *szl* expression as shown in the reported *szl* expression [16,37,38]. Bmp4 signaling responses and the spatio-temporal expression patterns indicated that the cloned *szl* promoter contained the expected Bmp4 regulatory elements required for synexpression.

### 2.2. Direct Smad1 Binding on Bmp4/Smad1 Response Cis-Acting Element (BRE) Site of Szl Promoter Is Essential for Bmp4 Synexpression

It has been known that for Bmp4 synexpression genes, including *szl*, *vent2*, *bambi* and *bmp7*, their transcription is regulated in a phospho-Smad1 dependent manner [13,39]; however, presence of a direct Smad1 regulatory response element or elements for the *szl* promoter has not been addressed. To determine whether Bmp4/Smad1 is directly involved in synexpression of *szl*, a protein synthesis inhibition experiment with cycloheximide (CHX) was performed. *Bmp4* mRNA (1 ng/embryo) was injected to the one-cell stage embryos and the animal caps were dissected at stage 8. The animal cap explants (ACs) were then cultured in L-15 media until stage 11 in the presence or the absence of CHX in order to determine whether *szl* is an immediate early (primary) or a secondary response gene to Bmp4. RT-PCR showed that the expressions of *szl* and a known Bmp4 direct target gene, *ventx2*, were maintained in the ACs treated with *Bmp4* mRNA in presence of CHX (Appendix A). By comparison, the expression of *xbrachyury (xbra)* disappeared in the CHX treated condition.

To determine the location of Bmp4/Smad1 response elements (BREs) within the *szl(-1566)* promoter, serially-deleted promoter constructs from the cloned *szl(-1566)* region were made (Figure 2a). The constructs were injected, with or without either *bmp4* or *smad1* mRNA, to one-cell stage embryos, and reporter gene assays were performed at stage 11. The relative reporter gene activities showed that five serially-deleted promoter constructs, including *szl(-1566)*, *(-1019)*, *(-716)*, *(-530)* and *(-370)*, were positively up-regulated by either Bmp4 or Smad1 (Figure 2b,c, bar 1–10). Ectopic expression of Bmp4 or Smad1 increased the relative luciferase activities by up to 8- to 10-fold when compared to those without *bmp4/smad1* mRNA co-injection (Figure 2b,c, bar 1–8). The shorter constructs, including three constructs *(-312) (-223)* and *(-197)*, did not respond to either Bmp4 or Smad1 ectopic expression (Figure 2b,c, bar 11–16), indicating that the putative cis-acting response element for Bmp4/Smad1 (BRE; Bmp4 response (or Smad1-binding) cis-acting elements) would be located between the *-370* and *-312 bps* locations of *szl* TLS. However, the Bmp4 inhibition (DNBR) response remained even in the smallest construct of *szl,* namely *szl(-197)* (Appendix A), suggesting that the downstream region of *szl(-197)* may contain negative regulatory element(s), which is addressed in Figure 3 and Figure 4. DNA sequence conservation between the *-370* and *-312 bps* of *Xenopus laevis* with that of *Xenopus tropicalis* indicated that the promoter region contained a putative conserved consensus sequence (indicated as BRE of *TCTG*; it is the reverse complement of *CAGA*; square red dotted box; Figure 2d). A *szl(-370)* promoter construct containing a mutated BRE from *-370* to *-312 bps* was made and then tested to determine whether the putative BRE was a functioning site for Smad1 (site-directed mutagenesis of BRE *TCTG* to *TTGG* with the mutated bases in red italics and underlined in Figure 2d). The wild type and mutated *szl(-370)mBRE* promoter constructs were injected at the one-cell stage and the reporter gene assays were performed at stage 11. To examine whether the putative BRE was the responding site for Smad1, the constructs with or without Smad1 overexpression were examined in reporter assays at stage 11 (Figure 2e). Smad1 increased relative luciferase activity of wild type *szl(-370)*, but it was completely abolished with the mutated *szl(-370)mBRE* construct. Direct binding of Smad1 on the 5′-flanking region of endogenous *szl* promoter was then examined for gastrula embryos with a quantitative ChIP-PCR experiment, using an anti-Flag antibody immunoprecipitation against injected *3Flag-Smad1* mRNA from the total chromatin of gastrula embryos. The results showed that Smad1 directly bound within the 5′-flanking region of the endogenous *szl* promoter (Figure 2f, 2nd lane) Together, the results showed that the *szl* promoter contains a direct Smad1 binding site (BRE) required for Bmp4 synexpression.

### 2.3. Two Direct Targets of Bmp4 Signaling, Ventx1.1 and Ventx2.1 Oppositely Modulate but Ventx1.1/2.1 Together Jointly Upregulate Szl Transcription

*Szl* transcription was positively regulated via direct binding of Smad1 on BRE site of *szl* promoter, in concordance with the expected results for *szl* synexpression with Bmp4, as Smad1 is a direct transcription factor mediating Bmp4 signaling. To determine the secondary response elements of Bmp4 signaling on the *szl* promoter, we examined the effects on *szl* transcription of two direct targets of Bmp4 signaling, the *ventx* genes *ventx1.1* and *ventx2.1.* Each promoter of these *ventx* genes contains a direct binding site of BRE, which is essential for their transcription in response to Bmp4 signaling [21,40]. To examine whether Ventx1.1 and Ventx2.1 transcription factors are involved in *szl* expression of whole embryos or not, *szl(-1566)eGFP* (*szl*-GFP) plasmid (40 pg/embryo) was injected at the one-cell stage with either *ventx1.1* or *ventx2.1* or *ventx1.1/ventx2.1* mRNA together (each at 0.5 ng/embryo). The embryos were then grown until stage 11 to detect their intensity of eGFP fluorescence signal. *Ventx1.1* co-injection decreased eGFP intensity (Figure 3a; bottom left panel). On the other hand, *ventx2.1* increased eGFP intensity on the ventral side of whole embryos (Figure 3a; Top right panel). Interestingly, the embryos co-injected with *ventx1.1/ventx2.1* mRNAs together showed slightly stronger fluorescence intensity than that of the *ventx2.1* alone injected group (Figure 3a; bottom right panel). To assess further effects of these Ventxs on *szl* transcription, *ventx1.1* and *ventx2.1* mRNAs were injected at the one-cell stage and the animal caps were dissected at stage 8. Quantitative RT-PCR analysis showed that the expression pattern of *szl* transcript was similar to those of eGFP fluorescence data from whole embryos. *Ventx1.1* mRNA injection decreased *szl* expression, and *ventx2.1* increased *szl* expression. *Ventx1.1/ventx2.1* co-injection showed more increased *szl* expression than that of *ventx2.1* alone injected ACs (Figure 3b). We observed similar results in whole embryos injected with the similar set of mRNAs through end point RT-PCR (Appendix A). To determine the location of Ventx1.1 inhibitory response element (VRE) within the *szl(-1566)* promoter, the serially-deleted *szl* promoter constructs were examined for response changes. The constructs were injected with or without *ventx1.1* mRNA (0.5 ng/embryo) to the one-cell stage embryos and reporter gene assays were performed at stage 11. The relative reporter gene activities showed that all eight serially-deleted promoter constructs of *szl(-1566)* to *(-197)* were more than five-fold down-regulated by *ventx1.1* (Figure 3c, bars 1–16). On a similar experiment, the relative reporter activities of all 8 *szl(-1566)* to *(-197)* constructs were about four-fold up-regulated by *ventx2.1* (Figure 3d, bars 1–16). The largest construct, *szl(-1566)luc*+, was then selected for examining the combination effects of *ventx1.1/ventx2.1*. As expected, *ventx1.1* injection decreased the relative reporter gene activity of *szl(-1566)luc*+ (Figure 3e; compare first (control) and second (*ventx1.1*) bars) and *ventx2.1* increased the reporter activity (Figure 3e; compare first (control) and third (*ventx2.1*) bars). Similar to the results of the eGFP and RT-PCR experiments, *ventx1.1/ventx2.1* mRNAs when together injected displayed the highest relative reporter gene activity (about seven-fold increase). *Ventx2.1* alone injected embryos showed a four-fold increase when compared to that of control samples (Figure 3e; compare first (control), third (*ventx2.1*) and fourth (*ventx1.1/ventx2.1*) bars). These data together indicated that the putative cis-acting response element(s) for both Ventx1.1 and Ventx2.1 (VRE(s); Ventx response cis-acting element(s)) may be located within the shortest *-197 bps* region of *szl* TLS. DNA sequence analysis was then performed to find a putative Ventx binding element within the *szl(-197)* region. We examined whether the promoter region contains the core consensus residues (*AAAT* or *ATTT*) between the *-196* and *-191 bps* of the proximal region of *szl* promoter. The *szl* promoter region contains a putative conserved consensus sequence (indicated as VRE, *TAAATT*, marked in square red dotted box; Figure 3f). The *szl(-370)* promoter construct containing the putative VRE site was mutated to identify whether the putative VRE was a functioning site for Ventx1.1 and/or Ventx2.1, or both. Taken together, the results showed that two direct target transcription factors (Ventx1.1 and Ventx2.1) of Bmp4 signaling oppositely modulate but together upregulate *szl* transcription, indicating that *szl* transcription is directly modulated by Bmp4/Smad1 signaling as well as by a secondary response using Bmp4 target transcription factors for Bmp4 synexpression.

### 2.4. Site-Directed Mutagenesis of One VRE Site on the Szl Promoter Abolishes Ventx1.1 as Well as Ventx2.1 Mediated Szl Transcriptional Regulation

DNA sequence analysis between the *-197* and *-1 bps* (Appendix A) region showed that the promoter region contained conserved consensus sequences that belong to a Ventx binding site (marked as VRE, Figure 3f). To identify whether the putative VRE were functioning sites for Ventx1.1 and/or Ventx2.1 or both, site-directed mutagenesis was performed on VRE (*TAAATT* to *TGGGTT*) (mutated bases, red italic and underlined; Figure 3f). Wild type *szl(-370)* or mutated *szl(-370)mVRE* promoter constructs were injected (40 pg/embryo) with and without *ventx1.1* or *ventx2.1* mRNA (0.5 ng/embryo) at the one-cell stage. The reporter gene assays were then performed at stage 11. For the mutated *szl(-370)mVRE*, the Ventx1.1-mediated as well as Ventx2.1- and Ventx1.1/Venx2.1-mediated relative luciferase activities of wild type *szl(-370)* construct were completely abolished (Figure 4a). The decreased response to Ventx1.1 and the increased responses to Ventx2.1 and Ventx1.1/Venx2.1 of wild-type *szl(-370)* had all disappeared in the single VRE mutated *szl(-370)mVRE* construct (Figure 4a, compare first to fourth bars with fifth to eighth bars). The results indicated that the same VRE was the responding site for Ventx1.1 and Ventx2.1. The direct binding of Ventx1.1 and Ventx2.1 on the VRE of endogenous *szl* promoter was then examined with stage 11 embryos. ChIP-PCR experiments were performed on Flag- or Myc-antibody immunoprecipitates from total chromatin for embryos injected with *3flag-ventx1.1* or *myc-ventx2.1* or the *3flag-ventx1.1/myc-ventx2.1* constructs. The ChIP-PCR and ChIP-qPCR results showed that Ventx1.1 and Ventx2.1 directly bound within the 5′-flanking region of endogenous *szl* promoter (Figure 4b, third lane of upper (Ventx1.1) and bottom (Ventx2.1) panel). In the co-injected *3flag-ventx1.1/myc-ventx2.1* samples, Ventx1.1 binding was reduced in presence of Ventx2.1 (Figure 4b, upper panel; compare third and fourth lanes) (Figure 4c; compare second and fourth bars). On the other hand, Ventx2.1 binding was enhanced by Ventx1.1 co-injection (Figure 4b, bottom panel; compare third and fourth lanes) (Figure 4c, compare third and fifth bars). To check the protein-protein interaction between Ventx1.1 and Ventx2.1, immunoprecipitation assay was performed at the gastrula stage of the embryo. 3F.Ventx1.1 and Myc.Ventx2.1 co-injected sample were pulled down with myc antibody, and through western blotting 3F.Ventx1.1 protein expression was detected, suggesting that Ventx1.1 and Ventx2.1 physically interact with each other and hence Ventx1.1 promotes the binding of Ventx2.1 on VRE site of *szl* promoter (Figure 4d). These results indicated that the Bmp4 primary as well as secondary response transcription factors Ventx1.1 and Ventx2.1 together are collectively participating in modulation of *szl* expression by binding to VRE of the *szl* promoter. For Ventx1.1/Ventx2.1-mediated regulation of *szl* transcription in a Bmp4 synexpression dependent manner, based on the results of Ventx2.1 and Ventx1.1 co-injection leading to a decreased amount of Ventx1.1 binding and an increased level of Ventx2.1 binding on VRE, the following schematic is proposed (Figure 4e). We hypothesize that Ventx2.1 binding would be assisted by DNA being unbound by Ventx1.1. However, as to which Ventx protein would bind first and how the two Ventx’s cooperate, these remain unknown. Together, these results suggest that one VRE site for Ventx1.1/Ventx2.1 binding is also involved in Bmp4 synexpression of *szl*.

### 2.5. Smad1 and Ventx1.1/Ventx2.1 Collectively Modulate Szl Transcription

The above findings show that Bmp4 signaling regulates the *szl* transcription via primary Bmp4 response mediated by Smad1 and a secondary Bmp4 response mediated by Ventx transcription factors. It has been known that the Smad1 (MH1) directly and specifically interacts with the C-terminal of Ventx2.1 in an additive fashion in transcriptional activation of Ventx2.1 [41,42,43,44]. Our results also showed a Ventx1.1/Ventx2.1-mediated positive regulation of *szl* transcription. To examine Smad1 and Ventx1.1 and/or Ventx2.1 combined effect on *szl* transcription, we injected *smad1* mRNA alone and in combination of *ventx1.1* and/or *ventx2.1* at the one-cell stage of *Xenopus* embryos. Animal caps were then dissected and maintained in L-15 medium. RNAs were isolated from the ACs and the qRT-PCR analysis was then performed. The results showed that *szl* mRNA expression was enhanced either by *smad1* or *ventx2.1* and reduced by *ventx1.1* (Figure 5a). As expected, *smad1* and *ventx2.1* showed a certain positive combination effect. However, *ventx1.1* retained the inhibitory response in *smad1* co-injected group shown as reducing the positive response of Smad1 mediated *szl* transcription. When *smad1* and *ventx2.1/ventx1.1* were injected together, *ventx1.1* inhibitory effect was not obvious and *szl* transcription was enhanced as much as that of *smad1* and *ventx2* co-injected groups. The reporter gene activity of *szl(-1566)* promoter construct was then measured after injection alone and with *smad1*, *ventx1.1* and *ventx2.1* mRNAs. The luciferase activity results were similar to those of quantitative RT-PCR (Figure 5b). To examine the difference in the amount of each protein bound on the *szl* promoter in the Smad1, Ventx1.1 and Ventx2.1 combined condition, ChIP-PCR analysis was performed with different sample sets for the combined injection groups of *ha.smad1*, *3flag.ventx1.1* and *myc.ventx2.1* mRNA-injected embryos at the gastrula stage. The results showed that Smad1 enhanced Ventx2.1 as well as Ventx1.1 binding on *szl* promoter (Figure 5c, seventh and ninth lane). On the other hand, Smad1 binding on *szl* promoter was decreased in presence of Ventx1.1 and increased in presence Ventx2.1 (Figure 5c, sixth and eight lane), indicating that Ventx1.1 inhibited the Smad1 binding on the *szl* promoter. Ventx2.1 and Smad1 combination enhanced each other’s binding on the *szl* promoter compared with singly present (Figure 5c, compare Smad1 binding (third (control) vs. Smad1/Ventx2.1 combination (ninth lane) and Ventx2.1 binding (first (control) vs. Smad1/Ventx2.1 combination (eighth lane). This finding is consistent with the previously reported data that Smad1 directly and specifically interacts with Ventx2.1 to activate *ventx2.1* transcription [43].

Smad1 and Ventx2.1 are positive regulators of *szl* transcription and are expected to cooperate with each other to enhance their binding on the *szl* promoter. Triple combination of Smad1, Ventx1.1/2.1 retained the reduced Ventx1.1 binding (Figure 5c, compare fifth and eleventh lane) found in the Ventx1.1/2.1 combination group. On the other hand, enhanced binding of each Smad1 and Ventx2.1 binding found in combination of Smad1 and Ventx2.1 was also retained in the triple combination group of Smad1, Ventx2.1/Ventx1.1 (Figure 5c, eighth and ninth vs. tenth and twelfth lane) (Figure 5d, ChIP-qPCR analysis). Smad1/Ventx1.1/Ventx2.1 triple action on *szl* expression is depicted as a schematic (Figure 5e), in which the interaction between Smad1 and Ventx2.1 is presumably maintained in the triple injection experiment including Ventx1.1 (Figure 5e). From Smad1 and Ventx2.1/Ventx1.1 triple injection results, these support a collective regulation for *szl* expression and maintenance of Smad1 and Ventx2.1 interaction (Figure 5b).

## 3. Discussion

In the present study, we aimed to uncover how Bmp4 synexpression is achieved by examining the promoter characteristics of *szl*, a Bmp4 synexpression gene. We found that the proximal region (within -370 from TLS) of the *szl* promoter contains cis-acting elements responding to Bmp4/Smad1 signaling as a BRE for a primary response. There was also another cis-acting element, one jointly responding to Ventx1.1 and Ventx2.1 as a VRE for a secondary response, confirming the binding of these two Bmp4 targets acting on the *szl* promoter. Interestingly, Ventx1.1 alone functions as a repressor (Figure 3a, bottom left panel; Figure 3b; Figure 3c, bars 1 to 16) when it binds on the VRE of the *szl* promoter, but functions as a coactivator enhancing the binding of the activator Ventx2.1 on the same VRE (Figure 4e). The implications of this study are briefly discussed below in light of the reported results.

### 3.1. Szl Is a Direct Target as Well as a Secondary Response Gene of Bmp4 Signaling

We demonstrated that *szl* is a direct target of Bmp4/Smad1 signaling in CHX experiments (Appendix A). Recently, Lee et al. also reported *szl* as being one of the direct targets of Bmp4/Smad1 signaling with in situ hybridization assays using CHX [26]. They also demonstrated the *szl* expression pattern in whole embryos using in situ hybridization [10]. In the present study, AC explants were used instead. Notice that the band intensity of *szl* in presence of CHX was weaker than that of untreated control in Bmp4 treated samples (Appendix A Appendix A, compare first vs. third and fourth lanes). We observed a similar pattern for *ventx1.1* expression, which is a primary as well as a secondary response gene for Bmp4/Smad1 signaling [21]. We speculate that synexpression is more effectively achievable when a gene is both a primary and secondary response gene of Bmp4/Smad1 signaling, since a small change in the morphogen gradient can switch the target gene expression via signal cascade amplification for secondary responses. It has been reported that *ventx2.1* expression is also dependent on both a primary and secondary response of Bmp4/Smad1 signaling through Smad1 and the C-terminal domain of Ventx2.1 [41,42,43,44]. Ventx1.1 expression is also positively modulated by *smad1*, *ventx2*, *gata2* and *xbra* [44,45]. It is necessary for the primary and secondary genes, including *smad1*, *ventx2.*1 and *ventx1.1* to be expressed at the same time and in the same cells during normal development. We examined temporal and spatial expression patterns of *ventx1.1*, *ventx2.1* and *szl*, as shown in Appendix A. RT-PCR results indicate that *ventx1.1*, *ventx2.1* and *szl* were expressed together for most of the relevant development stages (blastula to neurula) and on the ventral marginal region, but not in dorsal marginal region of normal *Xenopus laevis* embryos. At the present time, we do not know that Smad1, Ventx2.1 and Ventx1.1 were expressed in the same cells of *Xenopus* embryos. However, genes of both *ventx2.1* and *ventx1.1* are the direct targets of Bmp4/Smad1 [21] and are expressed in the ventral region (ventral mesoderm and ventral ectoderm) where *szl* is also expressed during blastula and gastrula *Xenopus* embryos. Synexpression genes of Bmp4/Smad1 signaling including *ventx2.1* and *ventx1.*1 may not be exclusively expressed in ventral cells of embryos at least until early gastrula stage since Ventx2.1 has been reported as an activator transcription factor on *ventx1.1* promoter working together with Smad1. The regulation differences and synexpression for various Bmp4 synexpression genes in the same cells remain to be investigated; these include the genes whose promoters have been cloned in the present study (*bmp7* and *bambi1*) (Appendix A). On modulation of *szl* expression, it also remains to be seen whether other secondary (and/or tertiary) transcription factors (TFs) including Gata2 and Xbra are also involved.

### 3.2. Ventx1.1 Enhances Ventx2.1’s Stimulatory Effect

One interesting observation of this study was the stimulatory transcriptional effect of Ventx1.1 when co-expressed with Ventx2.1 (Figure 4). Ventx1.1 is a well-known suppressive transcription factor and its repressor domain resides in its C-terminal domain [44,46]. Ventx1.1 primarily suppresses organizer gene expression and neural specific genes to protect the ventral mesoderm and ectoderm territory from being dorsal mesoderm and neuroectoderm, respectively [47,48,49]. Ventx1.1 also represses Bmp4 target genes including *gata2* [50], *ventx2.1* and itself [51]. The inhibitory function of Ventx1.1 was also shown for *ventx2.1* and *szl* (Figure 3b) in the present study. The direct and/or indirect suppressive effects of Ventx1.1 on certain individual genes need more elaboration, although some of the various Ventx1.1 inhibitory cis-acting elements in promoters have been reported [51,52]. In addition, Ventx1.1 competes with the transcriptional activator Xcad2 in binding the common cis-acting elements within the 5′-promoter region of *ventx1.1* to negatively regulate its own expression. [51]. In the present study, as a repressor, Ventx1.1 alone inhibits *szl* transcription. However, Ventx1.1 displays a stimulatory activity for *szl* expression and reporter activity as well as an increase in the amount of Ventx2.1 binding on VRE site in the *szl* promoter (according to co-injected group of Ventx1.1 and Ventx2.1). Consistent with this observation, Sander et al. reported that *ventx1.1*/*ventx2.1* double knockdown by morpholinos downregulates *szl* expression, indicating that *szl* is the downstream target gene of Ventx transcription factors [45]. Whang et al. also reported Ventx1.1 and Ventx2 physical binding using a yeast two-hybrid assay [44]. However, future work needs to determine whether the attenuation effect of Ventx2.1 on inhibition of Ventx1.1 is general to all Bmp4 synexpression target genes. Additional work needs to elucidate on how Ventx1.1 and Ventx2.1 physical co-binding on the promoter regions occurs with the resultant inhibition of Ventx1.1 suppressor activity. A more complicated picture is one for the Ventx1.1′s own promoter which contains two separate cis-acting elements, one for Ventx2.1 and one for Ventx1.1/Xcad2 competition [21,51]. The present study for the first time shows that Ventx1.1 and Ventx2.1 share one VRE site on the *szl* promoter and Ventx1.1/Ventx2.1 together positively modulate the expression of *szl,* a Bmp4 synexpression gene.

### 3.3. Smad1 and Ventx1.1/Ventx2.1 Collectively Upregulates Szl Expression

Kristine et al. reported that Smad1 directly and specifically interacts with Ventx2.1 and that Smad1 and Ventx2.1 co-injection synergistically activates the BRE on *ventx2.1* promoter [43]. Apart from Ventx2.1 as a target for Smad1/Ventx2.1, other target genes of this complex also need to be explored. Our results indicate that *szl* could be such a target of Smad1/Ventx2.1 complex, with *szl* mRNA expression and reporter activity assays, as well as Smad1 and Ventx2.1 binding all showing positive combination effects (Figure 5). However, the Ventx1.1 and Smad1 combination did not lead to a positive response in *szl* mRNA expression, reporter activity as well as Smad1 binding. Ventx1.1 retained the inhibitory response in *smad1* co-injection as shown by a reduction in the positive response of Smad1 mediated *szl* transcription, reporter activity (Figure 5b, compare second vs. sixth bar) and Smad1 binding (Figure 5c, compare third vs. sixth lane). When *smad1* and *ventx2.1/ventx1.1* were jointly injected, Ventx1.1 inhibitory effect seems to have disappeared as shown in the *ventx2.1/ventx1.1* combination group (Figure 5a), indicating that a triple combination of Smad1, Ventx1.1/2.1 retained the attenuation of Ventx1.1 inhibitory effect as shown in Ventx1.1/2.1 combination. Interestingly, Smad1 enhanced Ventx2.1 as well as Ventx1.1 binding on the *szl* promoter (Figure 5c, seventh and ninth lane). In addition, the enhanced binding of each Smad1 and Ventx2.1 found in Smad1/Ventx2.1 combination group was also retained in the Smad1/Ventx2.1/Ventx1.1 triple combination group (Figure 5c, eighth and ninth vs. tenth and twelfth lane).

Although additional Bmp4 synexpression targets of Smad/Ventx1.1/Ventx2.1 triple combination and their detailed underlying mechanism need to be explored, the present study provides an insight on how the Bmp4 synexpression pattern is achieved with the combinatory target gene effects, making up the primary and secondary responses of Bmp4 signaling and how an inhibitory transcription factor can become stimulatory under the right circumstances (Figure 6). However, it remains to be explored if removing any or all of the Bmp4 target proteins will affect the synexpression response patterns. Taken together, we propose a systematic model of Bmp4 signaling for regulation of *szl* transcription (Figure 6). This model elaborates the contribution of the Bmp4 response elements (Smad1 (BRE) and Ventx1.1/Ventx2.1 binding (VRE) sites) located on the *szl* promoter being responsible for Bmp4 synexpression. In summary, the *szl* promoter is collectively modulated by Smad1/Ventx1.1/Ventx2.1 for its Bmp4 synexpression in early *Xenopus* embryos.

## 4. Materials and Methods

### 4.1. Ethics Statement

This animal study was conducted in accordance with the regulations of the Institutional Animal Care and Use Committee (IACUC) of Hallym University (Hallym 2019-79, 2021-92). All the research members attended both the educational and training courses for the appropriate care and use of experimental animals at our institution in order to receive an animal use permit. Adult *X. laevis* were grown and tended to in approved containers; these were maintained at a 12 h light/dark (LD 12:12 h) cycle and 18 °C ambient temperature, by authorized personnel and according to the guidelines of the Institute of Laboratory Animal Resources of Hallym University for laboratory animal maintenance.

### 4.2. Nucleic Acids (DNA, RNA) Preparation

Various constructed cDNA were obtained by transformation of DH5alpha competent bacterial cells. The introduced cDNA clones including *pSP64T.bmp4*, *pSP64T.dnbr*, *3F.pCS4.smad1*, *3F.pCS4.ventx1.1* and *6Myc.pCS2.ventx2.1* were linearized with restriction enzymes EcoR1, Not1 and Acc65I and mRNA was made through in vitro transcription using a MEGA script kit (Ambion, Austin, TX, USA). All transcribed mRNAs were diluted in DEPC treated water and quantified at 260/280 nm wavelength using a spectrophotometer (SpectraMax, Molecular Devices, San Jose, CA, USA). Aliquots of 0.5, 1 ng/5 nL were made and stored at −80 °C for further use.

### 4.3. Cloning of Szl, Bmp7.1 and Bambi Promoter Constructs

Isolated genomic DNA from *Xenopus* blood were used for PCR amplification of *szl* promoter (−1566 bp), *bmp7.1* (−3143 bp) (accession number, NM_001087397.1) and *bambi* (−3073) (accession number, NM_001094009.1) having a Kpn1 restriction site at the forward and a Xho1 site at the reverse primer for the *szl* and *bmp7.1* promoters and having a Kpn1 restriction site at the forward and a Nhe1 at the reverse primer for *bambi* promoter. We amplified each promoter region by PCR cloning using the listed primers (Appendix A). The amplified *szl* promoter (−1566 bp) was subcloned into *pGL3.luc+* and *pGL3.eGFP+* (Promega, Madison, WI, USA) basic vectors. The cloned *szl* promoter (−1566 bp) was generated with the reverse primer sequence starting at −8 from TLS to guarantee including all the regulatory sequences for the transcription start of *szl* based on the deposited *szl*.L mRNA sequences (accession number; NM_001088521) which contains 5′UTR of 7 bps [53].

In addition, seven serially deleted constructs of various sizes were amplified from the −1566 bp fragment on the basis of mapping the potent Bmp signaling response elements and were cloned into *pGL3.luc+* and *pGL3.szl(−1566)eGFP+* constructs for fluorescence detection.

### 4.4. Xenopus Embryos Microinjection

Korea *Xenopus* Resource Center for Research was the source of adult *Xenopus laevis*. Five hundred units of human chorionic gonadotropin hormone (HCG) (Sigma, St. Louis, MO, USA) was injected to female *Xenopus leavis* in order to induce enough embryos. Freshly chopped male sperm was used for in vitro fertilization of the embryos in 1/20X MMR solution for 30 min. Embryos were de-gelled and subjected to various DNA and mRNA microinjections at the one-cell stage around the sperm entry point into the animal hemisphere. The injected embryos were grown and maintained in 30% MMR solution up to the required specific developmental stage for the experiments.

The *szl(−1566)eGFP* and *bambi(−3073)eGFP* constructs were injected (40 pg/embryo) with or without *dnbr* (1 ng), *ventx1.1*, *ventx2.1* mRNAs (0.5 ng each) per embryo at the one-cell stage and into the animal hemispheres. Green fluorescence was evaluated using a stereo microscope with royal blue light adapter (Stereo Microscope Fluorescence Adapter, NIGHTSEA, Hatfield, PA, USA), and photographs were captured using a Nikon D810 camera (Nikon, Minato City, Tokyo, Japan).

### 4.5. Embryos Animal Cap Explants Culture

Animal cap explants (10 AC/sample) of injected embryos were dissected at the early blastula stage, cultured in L-15 media in a 96-well plate, and harvested at the specific stage, followed by further experiments.

### 4.6. RT-PCR Experiment

Total transcribed RNA during a specific required stage was isolated from whole embryos using the TRIzol reagent (Ambion, Austin, TX, USA) and approximately 10 animal cap explants were used. The purified RNA was treated with DNase 1 to remove genomic DNA impurities. SuperScript IV (Invitrogen, Waltham, MA, USA) kit was used for cDNA preparation from 2 µg RNA for each sample.

### 4.7. Quantitative RT-PCR (qRT-PCR) Analysis

All RT PCR and Chromatin immunoprecipitation assays samples were subjected to qPCR analysis using Biosystems StepOnePlus Real-Time PCR System with Kapa Syber Fast qPCR Master mix. All values were normalized to ornithine decarboxylase (ODC), a house keeping gene of Xenopus embryos which is present all around the embryonic developmental. Graphs were made using threshold cycle (Ct) values, ΔCT, ΔΔCT and relative fold change expression method.

### 4.8. Luciferase Reporter Gene Assay

Embryos were injected with *szl* (−1566 bp) or its serially deleted constructs, mutated BRE and VRE constructs, or the *bmp7.1* and *bambi* constructs along with various indicated exogenous mRNA. Five sets of three embryos were collected at a specific stage and homogenized in lysis buffer (10 µL/embryo). Reporter gene activity was quantified using luciferase assay system (Promega) following the manufacturer’s instructions by luminometer (Berthold Technologies, Bad, Wildbad, Germany). Specific RT-PCR primers were used to amplify markers genes. (Appendix A).

### 4.9. Site Directed Mutagenesis of Smad1 and Ventx Response Elements (SRE and VRE)

Mutations for Smad1 response element (BRE) and Vent response element (VRE) sites were carried out by using the Muta-Direct site-directed mutagenesis kit (iNtRON Biotechnology, Seoul, Korea), following the manufacturer’s instructions. Mutagenesis PCR was performed from 40 pg/5 nl *szl* full-length promoter with specific mutagenesis primers (Appendix A).

### 4.10. Chromatin Immunoprecipitation (ChIP) Assay

Embryos were injected with *HA.pCS4.smad1*, *3F.pCS4.ventx1.1* and *6Myc.pCS2.ventX2.1* mRNAs at the one-cell stage either alone or together. Around 200 embryos/ sample at a specific stage were then collected following the reported protocol [54]. Anti-Flag (Sigma, F-1804) monoclonal antibody and anti-Myc (Santa Cruz, SC-789) (Santa Cruz, Dallas, TX, USA) polyclonal antibody were used for immunoprecipitation of crosslinked chromatin fragments. Mouse IgG (Santa Cruz, SC-2025) was used as negative control. ChIP-PCR was performed using specific primers (Appendix A).

### 4.11. Immunoprecipitation

Equal amounts of embryos were injected at one-cell stage with *3flag.ventx1.1* and *myc.ventx2.1* constructs mRNAs, separately and co-injected. All samples were collected at stage 11, and after removing MMR solution through spin down, embryos were homogenized in IP lysis buffer and followed the protocol as described in Kumar et al. (2018) [36]. The homogenized samples were centrifuged and the cleared lysates were then incubated with respective antibodies overnight at 4 °C. The immunocomplexes were precipitated by protein A/G beads (SC-2003, Santa Cruz Biotechnology, Santa Cruz, CA, USA). Proper amount of precipitated beads-immunocomplexes were heated up in the sample buffer and resolved by 10% SDS-polyacrylamide gel electrophoresis. Western blotting of 3Flag.Ventx1.1 and Myc.Ventx2.1 were performed by using Anti.Myc (Rabbit), Anti.Flag (Mouse) and secondary antibodies. Immune complexes were visualized with the help of ECL detection kit (GE Healthcare).

### 4.12. Statistical Analysis

GraphPad Prism 8 (GraphPad, San Diego, CA, USA) and MS Excel (Microsoft, Redmond, WA, USA) were used for plotting the graphs. Unpaired two-tailed Student’s *t*-test or ANOVA were applied for statistical analysis. *p* ≤ 0.05 for *, *p* ≤ 0.01 for **, *p* ≤ 0.001 for ***, *p* ≤ 0.0001 for ****, ns (not significant) were the assignments for significance.

## Figures and Tables

**Figure 1 ijms-23-13335-f001:**
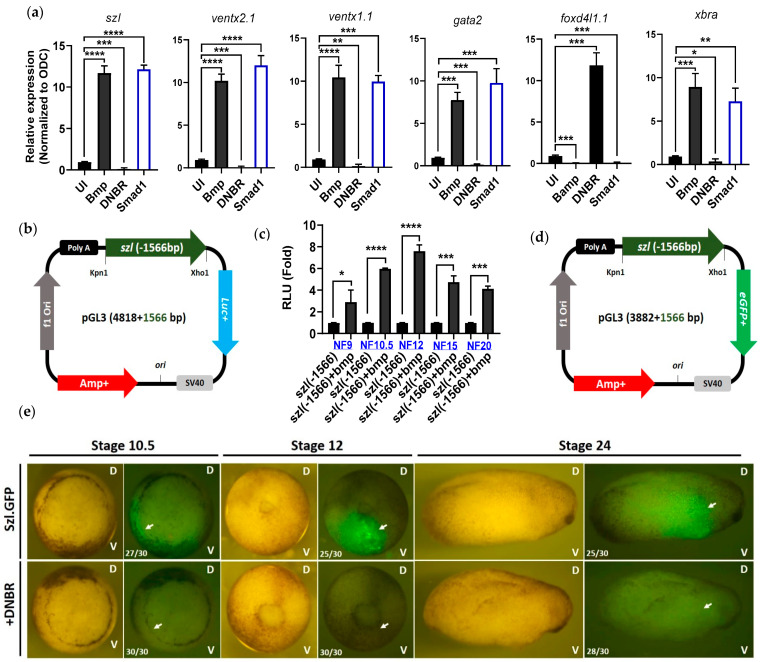
Bmp4 signaling modulates the isolated promoter activity of *szl(-1566bp)* in a stage-dependent manner: (**a**) *Xenopus leavis* embryos were injected with *bmp4*, dominant negative Bmp4 receptor (*dnbr*) and *smad1* mRNAs (1 ng/embryo) at the one-cell stage and animal caps (10 AC/sample) were dissected at stage 8 and grown up to stage 11. Ventral marker genes, including *szl* mRNA expression, were assayed through quantitative RT-PCR analysis. (UI: uninjected.) The qPCR values were normalized to ODC. (**b**) The *Szl(-1566)* promoter was cloned into *pGL3-luc+* vector. (**c**) Analysis of *szl* expression at various developmental stages in the presence of Bmp4, the reporter gene activity of the *szl(-1566)* promoter construct injected with *bmp4* mRNA was assayed. Five sets of three, three embryos were used for each sample in each developmental stage. (**a**,**c**) Unpaired two-tailed Student’s *t*-test or ANOVA were applied for statistical analysis. *p* ≤ 0.05 for *, *p* ≤ 0.01 for **, *p* ≤ 0.001 for ***, *p* ≤ 0.0001 for ****, ns (non-significant) were the assignments for significance. (**d**) *Szl(-1566)* promoter was cloned into the *pGL3-eGFP+* vector (*luc* was replaced with *eGFP*). (**e**) Embryos (30 embryos/sample) were injected with the reporter construct of *szl(-1566)-eGFP+* with or without *dnbr* mRNA and their eGFP fluorescence was observed at stages 10.5, 12 and 24. (D: dorsal, V: ventral.). The arrows indicate the regions of eGFP fluorescence appeared with *szl(-1566)-eGFP+* without *dnbr* mRNA and decreased with *dnbr* mRNA.

**Figure 2 ijms-23-13335-f002:**
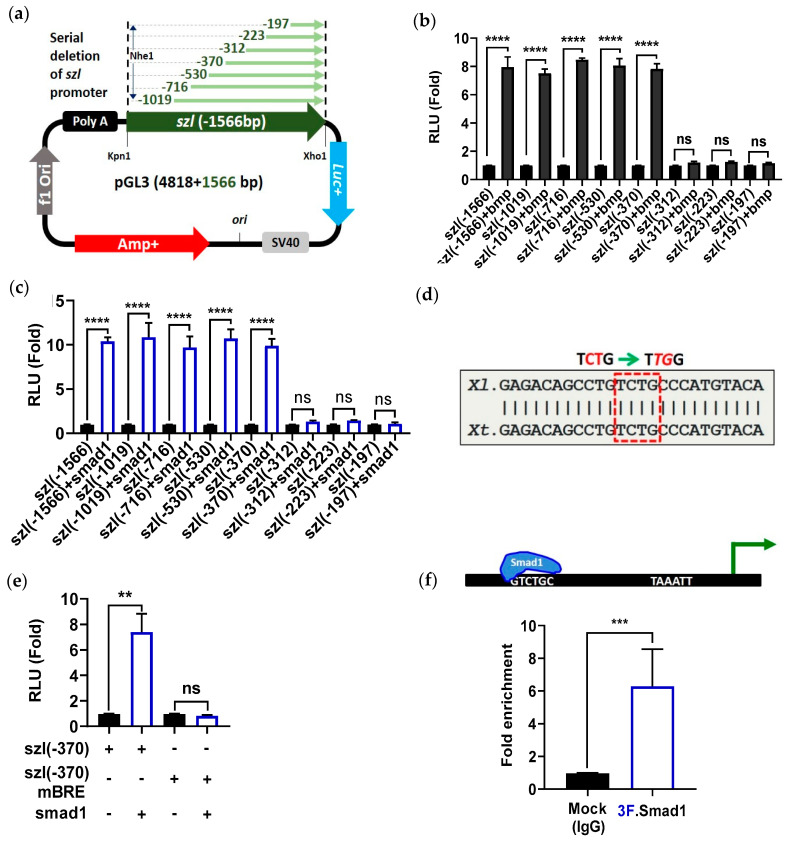
Bmp4 signaling upregulates *szl* reporter gene activity via Smad1 regulatory response elements within the *szl* promoter: (**a**) Schematic diagram of serially deleted constructs of *szl* promoter. (**b**) Reporter gene activities of the serially deleted constructs of *szl* promoter injected with or without *bmp4* and (**c**) *smad1* mRNAs. (**d**) Conserved region of *szl* promoter consisting of a Bmp4 response element (BRE) (outlined within the dashed red box). (**e**) Luciferase reporter gene activity of *szl(-370)* wild type and mutated BRE constructs injected with or without *smad1* mRNA were examined. (**b**,**c**,**e**) *p* ≤ 0.01 for **, *p* ≤ 0.001 for ***, *p* ≤ 0.0001 for ****, ns (non-significant) were the assignments for significance. (**f**) Embryos were injected with *3flag-smad1* mRNA and quantitative ChIP-PCR was performed with anti-Flag immunoprecipitated genomic DNA using specific primers for the *szl* promoter fragment having the Smad1 binding site. Fold enrichment method used to normalize ChIP-qPCR. Smad1 binding scheme is given in the upper part of figure. (SRE: Smad1 response elements).

**Figure 3 ijms-23-13335-f003:**
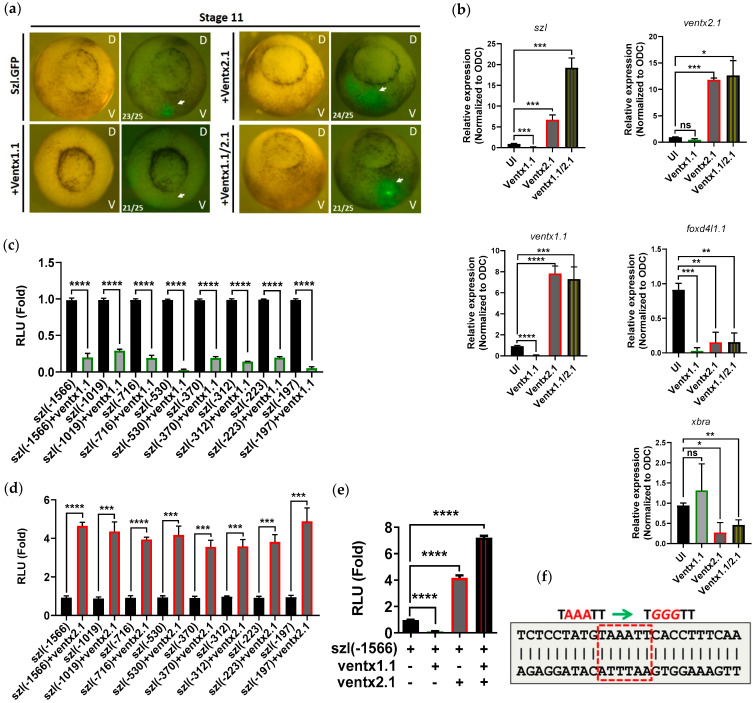
Ventx1.1 and Ventx2.1 mediated *szl* transcriptional modulation acts as a secondary Bmp4 signaling response: (**a**) Green fluorescence of embryos injected with *szl(-1566)-eGFP+* in combination of *ventx1.1* and/or *ventx2.1* mRNAs at the one-cell stage was observed at stage 11. (25 embryos were injected per sample) (**b**) Embryos were injected with, without or with some combination of *ventx1.1/ventx2.1* mRNAs. Animal cap explants (10 AC/sample) were dissected at stage eight and relative expression of ventral specific marker genes, including *szl* at gastrula stage, through quantitative RT-PCR was performed. Ct values were normalized to ODC, a housekeeping gene. (**c**) Serially deleted constructs of the *szl* promoter were injected with or without *ventx1.1* and (**d**) *ventx2.1* mRNAs. Reporter gene activities were analyzed. (**e**) *Szl (-1566)*-expressing embryos were injected with, without or with some combination of *ventx1.1/ventx2.1* mRNAs and analyzed for luciferase activity. (**b**–**e**) *p* ≤ 0.05 for *, *p* ≤ 0.01 for **, *p* ≤ 0.001 for ***, *p* ≤ 0.0001 for ****, ns (non-significant) were the assignments for significance. (**f**) Fragment of *szl* promoter having the putative Ventx response elements (VRE, highlighted within the dashed red box).

**Figure 4 ijms-23-13335-f004:**
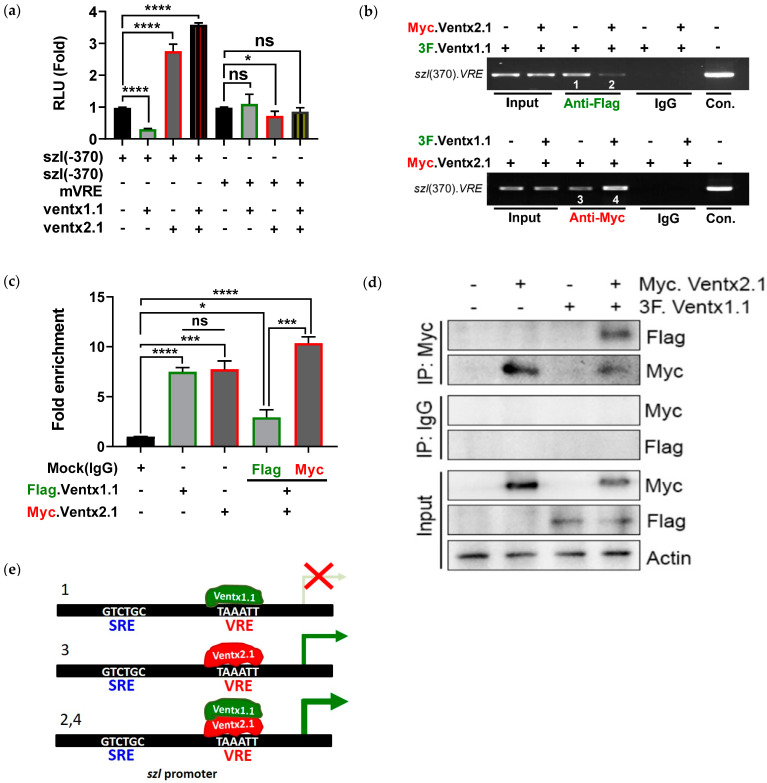
One VRE (*ventx* cis-acting response element) site for Ventx1.1/2.1 binding in *szl* promoter is required for *szl* transcription: (**a**) *Szl(-370)* wild type or *szl(-370)*mVRE were injected with the indicated mRNAs and their combination. Luciferase activities of the samples were analyzed. (**b**) Embryos (300 embryos/sample) were injected alone or with the combination of *3flag-ventx1.1* and *myc-ventx2.1* mRNAs (1 ng/embryo) and cultured to the gastrula stage. ChIP-PCR analysis using specific primers for *szl* promoter region having the VRE site was performed. (**c**) Graph represents the relative band intensities of ChIP-PCR bands. (**a**,**c**) *p* ≤ 0.05 for *, *p* ≤ 0.001 for ***, *p* ≤ 0.0001 for ****, ns (non-significant) were the assignments for significance. (**d**) Immunoprecipitation assay were performed for 3F.Ventx1.1 and Myc.Ventx2.1 protein-protein interaction during *szl* promoter binding. (30 embryos/sample) (**e**) A model of Ventx1.1/2.1 alone and combination mediated regulation of *szl* transcription. The numbers indicate the number of lane shown on (**e**), indicating that each schematic model diagram is related with and based on the result(s) of lane(s) shown on (**b**).

**Figure 5 ijms-23-13335-f005:**
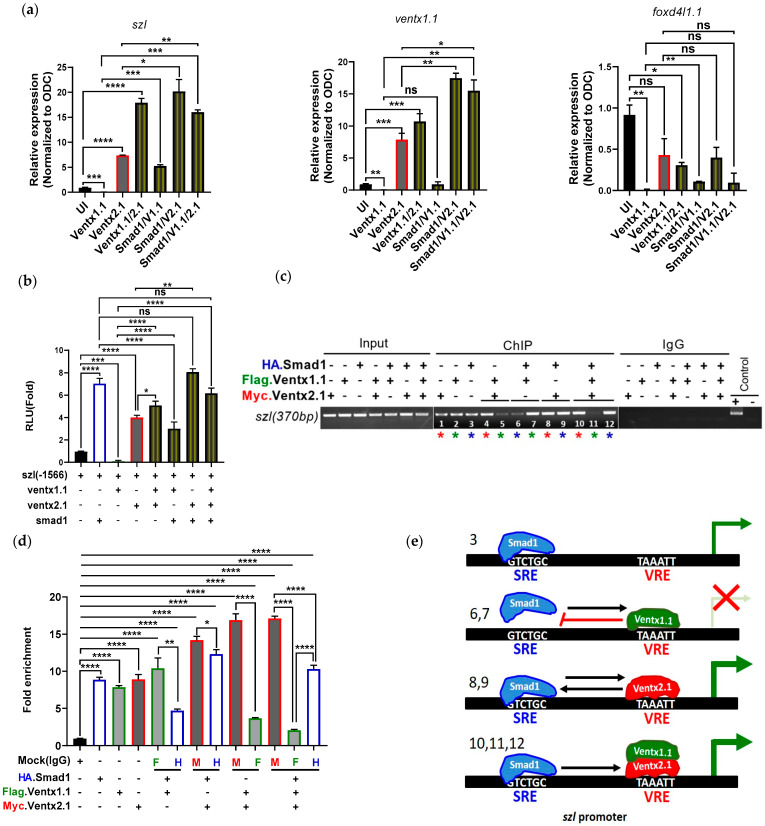
Smad1 enhances Ventx2.1-mediated *szl* activation and rescues Ventx1.1 mediated *szl* suppression. (**a**) Embryos were injected with *smad1*, *ventx1.1* and *ventx2.1* mRNAs individually and in different combinations. Animal cap explants (10 AC/sample) were dissected at stage eight and quantitative RT-PCR of ventral specific marker genes including *szl* at gastrula stage were analyzed. V1.1 = Ventx1.1, V2.1 = Ventx2.1 (**b**) *Szl(-1566)* promoter construct were injected with *smad1*, *ventx1.1* and *ventx2.1* mRNA alone and possible combinations and reporter gene activities were analyzed. (**c**) *Xenopus leavis* embryos (300 embryos/sample) were injected with *ha.smad1*, *3flag.ventx1.1*, *myc.ventx2.1* mRNAs alone and their possible combinations (0.5 ng/embryo each) and cultured to the gastrula stage. ChIP-PCR analysis using specific primers for the *szl* promoter region having the VRE site was performed. The asterisks of different colors indicate the difference of antibodies used for ChIP; red (*****) for Myc.Ventx2.1, green (*****) for Flag.Ventx1.1 and blue (*****) for HA.Smad1. (**d**) Quantitative PCR analysis was performed with fold enrichment method and the experiment was repeated three times. (**a**,**b**,**d**) *p* ≤ 0.05 for *, *p* ≤ 0.01 for **, *p* ≤ 0.001 for ***, *p* ≤ 0.0001 for ****, ns (non-significant) were the assignments for significance. (**e**) A model of Smad1 and Ventx1.1/2.1 alone and in combinations mediated regulation of *szl* transcription. The numbers indicate the number of lane shown on (**c**), indicating that each schematic model diagram is related with and based on the result(s) of lane(s) shown on (**c**).

**Figure 6 ijms-23-13335-f006:**
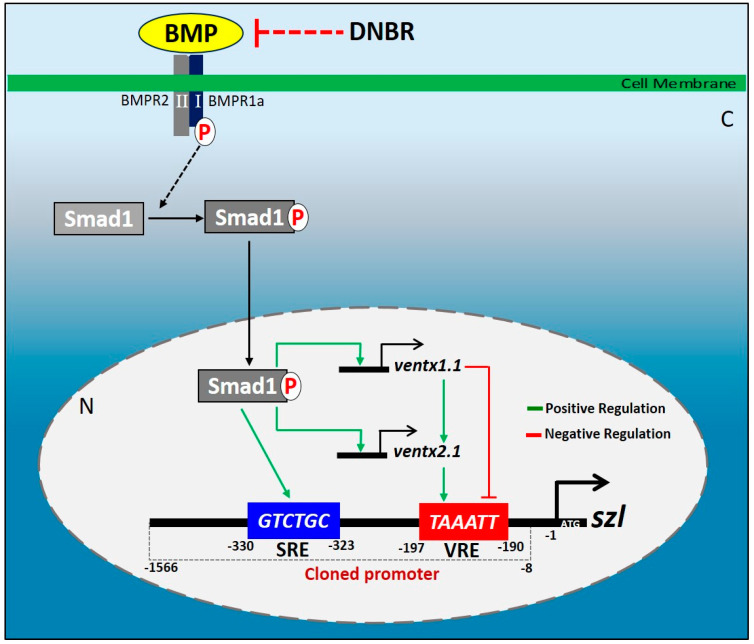
Proposed model of the *szl* transcriptional regulation by Bmp4 signaling via Smad1/Ventx1.1/Ventx2.1 axis. This schematic overview of *szl* transcriptional regulation during Bmp4 synexpression fashion is based on our findings above. *Szl* promoter has two types of Bmp4 response elements required for synexpression; they are the Smad1 binding response element (SRE) and the Ventx binding response element (VRE). Smad1 binds to SRE (reported here as *TCTG*) while Ventx1.1 and Ventx2.1 bind to one VRE (described herein as *TAAATT*) to regulate *szl* expression. Ventx1.1 reduces *szl* expression, while Ventx2.1 induces it, and its combination further enhances it, compared to Ventx2.1 presence alone.

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
