# Peer review of "Bmp4 Synexpression Gene, *Sizzled,* Transcription Is Collectively Modulated by Smad1 and Ventx1.1/Ventx2.1 in Early *Xenopus* Embryos"

_ijms, 2022, doi:10.3390/ijms232113335_

Round 1
Reviewer 1 Report
This paper examines the transcription factors that regulate the expression of the sizzled (szl) gene. Szl has a pattern of expression in the Xenopus embryo that is very similar to BMP4 and is considered part of the BMP4 defined synexpression group. Identifying the transcription factors that regulate szl expression will help explain how the synexpression group is established.
Overall the experiments are clear and informative and indicate that szl expression is regulated by Smad1 and by a combination of Ventx2.1 and Ventx1.1 transcription factors, allowing the construction of a potential simple gene regulatory network. This is a solid piece of work that provides reliable detail on this system.
I have some minor comments:
1. The 1566bp 5’ flanking region used to map the promoter/enhancer sites begins at the translational start site of szl. The authors could comment on the extent of the (short) 5’UTR in szl.
2. Although the RT-pcr analysis is generally convincing, it would have been more appropriate to use a quantitative (qPCR) approach. The authors should comment on the limitations of this semi-quantitative approach.
3. On line 300 and later, the authors refer to changes in the binding affinity of transcription factors on the szl promoter. This is probably not the correct term. Biochemically, binding affinity is a quantitative value measured as a Kd. This level of biochemical data cannot be obtained from the ChIP experiments reported in the paper. I agree that there is a difference in the amount of protein bound in the ChIP assays but this needs to be expressed more accurately in the paper.
4. One aspect of the paper is the interaction of Ventx2.1 and Ventx1.1. It would be useful if the authors could comment on the evidence of whether these two factors are expressed at the same time in the same cells during normal development.
Reviewer 2 Report
The manuscripts is a comprehensive report, analysing the cis-regulatory elements that control the expression of Sizzled in response to BMP4 signalling. The authors present a logical series of experiments, from determining the response of Szl reporters to BMP4, narrowing the cis-regulatory element by deletion analysis, mutating the binding sites to ChIP experiments for transcription factor binding. The results suggest a dual control for Szl expression, a primary response mediated by Smad1 and a secondary response mediated by Ventx2.1, which is a novel finding. Also interesting is the binding of Ventx1.1 and Ventx2.1 to the same response element, with opposite effects on Szl expression, and strong indication that Ventx1.1 binding enhances the subsequent binding of Ventx2.1.
While the findings are interesting and the results principally look solid, some improvements sould be made.
The n numbers are missing for any of the experiments. These should be included in the figure legends.
It is disappointing that all RT-PCR and ChIP-PCR experiments were performed as endpoint PCR rather than qPCR. While the results look principally convincing, this approach ruled out the opportunity for quantifiable results. To some extent the authors compensated for the lack of qPCR expression data by utilising the luciferase assays, but it still limits the significance of the study, particularly since qPCR (at least SYBR Green) is nowadays considered a standard procedure.
Some more specific comments:
There is a discrepancy in the RT-PCR data for Szl in Fig. 1a/b: in Fig. 1a, Szl expression in the uninjected animal cap is much higher than in whole embryo, while Fig. 1b shows the opposite. This is in contrast to the consistent results for Ventx1.1 and Xbra, and needs to be discussed.
It might be better to show both reporter constructs as Fig. 1c (or indicate in the figure that the reporter gene was either luciferase or GFP), and then have the results of the reporter assays as Fig. 1 d/e.
The resolution of Fig. 1e is very poor (probably a result of embedding it into the PDF file). It is difficult to assess the similarity of the reporter gene expression with endogenous Szl expression without showing the Szl expression pattern – showing the in situ hybridisation pattern for the gene alongside the GFP fluorescence would be helpful. The effect of the dnBR injection is convincing, though.
Fig. 3f/lines 242-244 should be moved to part 2.4 (the results for the reporter with the mutated VRE); both the figure and the sentence only make sense together with the results.
Clever use of the myc- and FLAG-tagged constructs in Fig. 4b. Would be great to have qPCR to quantify the ChIP result, rather than the less reliable semi-quantitative end-point PCR.
In Fig. 4d it is not clear what mechanism for the joint binding the authors propose. It seems to suggest that Ventx1.1 somehow interacts with VRE-bound Ventx2.1, but the evidence for, or result of such interaction is not explained. The results would rather indicate initial binding of Ventx1.1, which might promote the (higher affinity?) binding of Ventx2.1. A minor point about Fig. 5e: the interaction between Smad1 and Ventx2.1 is presumably maintained in the triple injection experiment, considering the reporter data.
Specific text corrections and suggestions:
Line 68: signalling, directly controlling (delete comma)
Line 103: produces dorsal organizer and neural genes (sloppy phrasing; rather: ‘results in upregulated expression of ‘)
Fig. 1 legend: Embryos were injected with or without dnbr mRNA the reporter (muddled; rather: Embryos were injected with the reporter mRNA with or without dnbr mRNA…)
Lines 137-139: RNA expression, eGFP imaging and reporter gene activities of szl in various developmental stages indeed indicated a similar pattern of expression. (does this refer to the results shown in Fig. 1e? If yes, this should be stated)
Line 198: szl transcription on two direct targets of Bmp4 signaling (presumably this should be ‘of’, since the Ventx genes affect Szl transcription, not the other way round)
Line 348: Although, it has been reported that (delete comma)
Line 354: which whose promoters hasve been cloned in the present study
Line 363: its repressor domain resides in the its C-terminal domain
Line 369: effects of Ventx1.1 to on certain individual genes
Line 371/372: Ventx1.1 competes one cis-acting element site with a transcriptional activator Xcad2 to regulate negatively its own expression (unclear; do you mean: ‘Ventx1 competes with the transcriptional activator XCad2 in binding a cis-acting element to negatively regulate its own expression’?)
Line 392: They reported that Smad1 and Ventx2.1 coinjection synergistically activates the
Round 2
Reviewer 2 Report
The authors have addressed all my comments. I am particularly please that the end point PCR data have been replaced by qPCR data that not only are consistent with the previous findings, but provide much stronger support for the findings.
Reviewer 3 Report
The authors addressed all the major concerns and improved the quantity of the paper with additional data, proper controls, analysis, and explanatory texts.
Some of the sentences are long and complicated. Minor English editing will help improve the reading experience.